# PTSD symptoms among health workers and public service providers during the COVID-19 outbreak

**Sverre Urnes Johnson**[1,2]*, **Omid V. Ebrahimi**[1,2], **Asle Hoffart**[1,2]

**1** Department of Psychology, University of Oslo, Oslo, Norway, **2** Modum Bad Psychiatric Hospital, Vikersund, Norway

☯ These authors contributed equally to this work.

\* s.u.johnson@psykologi.uio.no

**Data Availability Statement:** Our ethical approval granted by the Regional Committees for Medical and Health Research Ethics in Norway does not allow us to submit the data to a Public repository. In line with the ethics approval, the data are to be

## Abstract

In the frontline of the pandemic stand healthcare workers and public service providers, occupations which have proven to be associated with increased mental health problems during pandemic crises. This cross-sectional, survey-based study collected data from 1773 healthcare workers and public service providers throughout Norway between March 31, 2020 and April 7, 2020, which encompasses a timeframe where all non-pharmacological interventions (NPIs) were held constant. Post-traumatic stress disorder (PTSD), anxiety and depression were assessed by the Norwegian version of the PTSD checklist (PCL-5), General Anxiety Disorder −7, and Patient Health Questionnaire-9 (PHQ-9), respectively. Health anxiety and specific predictors were assessed with specific items. Multiple regression analysis was used for predictor analysis. A total of 28.9% of the sample had clinical or subclinical symptoms of PTSD, and 21.2% and 20.5% were above the established cut-offs for anxiety and depression. Those working directly in contrast to indirectly with COVID-19 patients had significantly higher PTSD symptoms. Worries about job and economy, negative metacognitions, burnout, health anxiety and emotional support were significantly associated with PTSD symptoms, after controlling for demographic variables and psychological symptoms. Health workers and public service providers are experiencing high levels of PTSD symptoms, anxiety and depression during the COVID-19 pandemic. Health workers working directly with COVID-19 patients have significantly higher levels of PTSD symptoms and depression compared to those working indirectly. Appropriate action to monitor and reduce PTSD, anxiety, and depression among these groups of individuals working in the frontline of pandemic with crucial societal roles should be taken immediately.

## Introduction

The psychological and social consequences of the COVID-19 pandemic have a pervasive effect on current mental health [1–3]. In the frontline of the pandemic stand healthcare workers and public service providers, occupations which have proven to be associated with increased mental health problems during pandemic crises [4, 5]. In particular, these workers are vulnerable

kept at a secure server only accessible by the authors at the University of Oslo. Access to the data can be granted from the first author following ethical approval of suggested project plan for the use of data from NSD and REK. Such requests are to be sent to Associate Professor, Sverre Urnes Johnson, Department of Psychology, University of Oslo, Forskningsveien 3 A, 0373 Oslo, Norway, Email: s.u.johnson@psykologi.uio.no, phone: +47-22845295, or to Omid V Ebrahimi, Email: omid.ebrahimi@psykologi.uio.no. The data are stored at the TSD-system, which is part of the long-term storage facility at the University of Oslo. TSD uses regular back-up thus the data is well secured.

**Funding:** The author(s) received no specific funding for this work.

**Competing interests:** The authors have declared that no competing interests exist.

to developing post-traumatic stress disorder (PTSD) symptoms. PTSD is a mental health problem that affects people who are exposed to potentially traumatic episodes. Healthcare workers are exposed to increased danger of contamination, loss of patients, responsibility for difficult decisions on treatment retention, and disruption of normal supportive structures [1, 6].

PTSD symptoms are grouped into 4 clusters: re-experiencing, avoidance, negative cognitions and mood, and arousal, according to the Diagnostic and Statistical Manual of Mental Disorder, Fifth Edition (DSM-5). Most studies on PTSD report lifetime prevalence, which gives higher estimates of prevalence compared to point prevalence. In American and Canadian studies, with samples from the general adult population, the lifetime prevalence varies from 6.1% to 9.2% [7–10]. Estimates were, however, lower in a World Health Organization study reporting a lifetime prevalence of PTSD in upper-middle and lower-middle income countries of 2.3% and 2.1% [10].

The estimates of PTSD symptoms among healthcare workers are higher compared to the general population and range from 6–10% in a recent COVID-19 survey conducted in Singapore [11], 18% from nurses working in hospitals in general [12], and 20% from the Severe Acute Respiratory Syndrome (SARS) outbreak [13]. Thus, PTSD symptoms appear to be higher during pandemics compared to periods without extraordinary situations.

As reported in earlier pandemics including SARS and Middle East Respiratory Syndrome (MERS), working directly with infected persons has been associated with high levels of PTSD symptoms [14–16] and worry [17]. Thus, evidence from previous studies indicates that working closer and more intensively with COVID-19 patients is associated with higher mental distress.

What predicts the development of PTSD symptoms is another critical question. Meta-analyses [18, 19] and reviews [20, 21] have found that sociodemographic (i.e., female gender and young age) [18, 19], prior mental disorder such as anxiety and mood disorder [20, 21] and social support are associated with PTSD symptoms [18]. Other important predictors found in previous studies include worry [22, 23], burnout [24], interpersonal problems [25], and positive and negative metacognitive beliefs [26, 27]. Positive metacognitive beliefs exist about the usefulness of worry, rumination, threat monitoring and other coping strategies (i.e., "If I worry I will be prepared"). Negative metacognitive beliefs concern the uncontrollability of thoughts and perceived danger (i.e., "I cannot control my thoughts"). Positive and negative metacognitive beliefs give rise to a specific way of tackling emotional distress characterized by worry and rumination, which prolongs and intensifies distress [28].

Thus, a variety of possible predictors of PTSD symptoms exist, including static (sociodemographic) and state predictors, some of which are central to the treatment of PTSD (interpersonal problems, worry and metacognitions).

Research into the mental health consequences for frontline workers in the current ongoing pandemic is critically needed, especially public service providers, providing the basis for the development of adequate treatment and possible prevention of mental problems during the present as well as future pandemics, as recently reflected by multiple urgent calls in the literature [1–3, 29]. To date, no examination differentiating between individuals working directly with infected patients (health personnel) versus professionals working indirectly with pandemic consequences (politicians, social workers and other health personnel) has been conducted in any pandemic, leaving gaps in the literature concerning how these divergent groups differ in mental health outcomes during pandemics. This gap is crucial to fill, as it refines understanding of how different vulnerable professionals working directly versus indirect with pandemic consequences are impacted psychologically, providing the foundation for forthcoming interventions aimed at reducing these symptoms. As a pandemic involves a widespread burden on different labor forces with divergent impacts on each labor group given the tasks of

different occupations, an investigation revealing the psychopathology levels of these under-studied work forces will be imperative in order to protect them against detrimental mental health outcomes.

The present study aims to provide an assessment of the mental health burden of healthcare workers and public service providers working directly and indirectly with those infected with the COVID-19 virus. Directly is defined as face-to-face contact with patients that have tested positive for COVID-19. Indirectly is defined as working with other consequences of the COVID-19 pandemic, but not face-to-face with patients that is infected with COVID-19.

The following hypotheses were formulated:

H1: Health employees working directly with COVID-19 victims will have higher symptoms of PTSD, anxiety, depression, and health anxiety, compared to health workers and public service providers working indirectly with the virus.

Exploratory: Examine the differences in levels of PTSD symptoms, anxiety, depression and health anxiety among different subgroups of health workers and public service providers.

H2: Presence of a psychological disorder and more anxiety and depression symptoms will be associated with more PTSD symptoms.

H3: Less emotional support, more burnout, more health anxiety, more worries about job and economy, more interpersonal problems and stronger metacognitions will be associated with more PTSD symptoms controlling for direct vs. indirect exposure to trauma, pre-existing psychiatric diagnosis, anxiety and depression, and demographic variables (age and gender, living with a partner, living with children).

## Materials and methods

### Study design and recruitment procedure

The study has a cross-sectional survey design. Health personnel and public service providers were systematically targeted through various channels: First, a selection of those qualifying as healthcare workers and public service providers were randomly targeted on Facebook. The Facebook algorithm reaches a random sample of individuals including and above 18 years of age who have reported their labor to fit our target categories, such as nurses, doctors, psychologists, and other individuals working in the health-care system, in addition to those reported to be politicians and social workers. The algorithm is inherently designed to optimize for genuine human activity and uses a variety of methods to remove false and duplicate accounts from the selection algorithm. Upon taking the survey, these participants identify and register themselves through a platform referred to as Services for Sensitive Data (TSD), where their information is safely stored. With the imputed parameters, the algorithm reached a total of 12 113 individuals meeting the aforementioned criteria. Second, hospitals in Norway were reached out to systematically and health personnel were invited to participate. Third, the associations of all major health worker groups were contacted. Moreover, national TV, national, regional and local radio stations, and national, regional and local newspapers were used. Politicians were further systematically contacted, with all political parties sending an e-mail with the survey to their members. Participants were asked to fill out a set of validated questionnaires including demographic variables, psychological symptoms including symptoms of depression, anxiety and PTSD symptoms.

The period of data collection lasted seven days and was undertaken between March 31, 2020 and April 7, 2020, which encompasses a timeframe where all non-pharmacological interventions (NPIs) were held constant during the two weeks prior to data collection, as well as during the data collection week. NPIs are actions that people, and communities can take to help slow the spread pandemics, like the COVID-19. No information was given from the

government about possible changes in any epidemic protocols during the data-collection, controlling for expectation effects.

Approval from the Regional Committee for Medical Research Ethics was received prior to commencement of study (reference number: 125510). Participants were allowed to terminate the survey at any time without any consequences. The study was pre-registered at Clinical-Trials.gov after data collection, but before any analysis (Identifier: NCT04374097) and is part of the Norwegian COVID-19, Mental Health and Adherence project. Articles from the same project, but with different topics, concerning the prevalence of anxiety and depression [30], loneliness [31] and parental stress [32] are under consideration for publication.

## Participants

Participants eligible for participation were individuals > 18 years of age, who have provided their consent to participate in the survey. The participants were either health personnel or public service providers working directly or indirectly with COVID-19 patients. The following groups of health personnel were assessed: Medical doctors, nurses, clinical psychologist and other health workers (not specified). The following group of public service providers working with consequences of the COVID-19 pandemic were assessed and grouped: social workers, politicians and other professions (not specified).

The target sample size of 1900 participants was determined using a conservative recommendation of a sample size ten times larger than the estimated parameters for multivariate analysis [33].

## Outcomes and covariates

The following measure were used in the current study:

*PTSD checklist for DSM-5 (PCL-5)* were used to measure PTSD symptoms (0–80), which had an $\alpha = 0.94$ in the current sample [34]. The participants were asked about the most distressing event from the COVID-19 period. The DSM-5 diagnostic guidelines were applied to the PCL-5 to categorize participants as fulfilling the PTSD symptom criteria or not [34].

Participants indicating scores of 2 or above on at least one of five re-experiencing symptoms, one of two avoidance symptoms, two of seven symptoms of negative alterations in cognition and mood and two of six arousal symptoms were classified as fulfilling the PTSD symptom criteria (maximum of 6). Subclinical PTSD was defined as those who had at least 4 criteria fulfilled, with a diagnostic score in at least two symptom clusters of PTSD. This is in accordance with the recommendation in the literature of two to three DSM-5 criteria in symptom clusters fulfilled [35].

*General Anxiety Disorder -7 (GAD-7)* was used to measure anxiety (0–21) and had a Cronbach $\alpha = 0.87$ in this sample [36]. *Patient Health Questionnaire-9 (PHQ-9)* was used to measure depression (0–27), $\alpha = 0.86$ [37]. For GAD-7 the scores were classified as; normal (0–4), mild (5–7), moderate (8–14) and severe (15–21). For the PHQ-9 scores; normal (0–4), mild (5–9), moderate (10–14) and moderately severe (15–19) and severe (20–27). These cut-offs have been well established in the literature [36–38].

Four items were combined into the subscale positive metacognition, $\alpha = 0.62$, and four items were combined into the subscale negative metacognition, $\alpha = 0.69$, from the Cognitive Attentional Syndrome Scale-1 (CAS-1) [39]. Seventeen items from the Inventory of interpersonal problems (IIP) [40] were combined into the scale interpersonal problems, $\alpha = 0.81$. Four variables that measure health anxiety and fear of death related to COVID-19 represented health anxiety, $\alpha = 0.77$. Furthermore, two items were combined to represent worries about work and economy, $\alpha = 0.72$. Three items were combined to measure emotional support, $\alpha =$

0.79. Burnout was measured through a single item, "I feel burned out". See S1 File for an overview of all the different items. The demographic data assessed were gender (male, female, transgender), type of occupation (doctors, nurses, clinical psychologists, social workers, politicians and other health workers), age (18–24, 25–44, 45–59, >60), marital status, living with children and education level.

### Statistical methods

There were no missing data, because the online survey system included mandatory fields of response. However, in analyses involving gender, transgendered (N = 1) and intersexed individuals (N = 1), there were too few individuals to be included. First, as the variables were highly left skewed, the level of PTSD symptoms, anxiety, depression, and health anxiety between those that worked directly with Covid-19 patients and indirectly was compared using the Mann-Whitney U test. Second, the different groups was compared using the Kruskal-Wallis test. Third, two multiple regression analyses with PTSD symptoms as the dependent variable was performed. The first with anxiety, depression, and pre-existing psychiatric diagnosis as predictors. The second assessed predictors of trauma symptoms such as worry, health anxiety, burnout, emotional support, interpersonal problems and positive and negative metacognitions, after controlling for confounders.

In all regression analyses, multicollinearity and other assumptions were checked; in particular if the multicollinearity assumption was violated (if VIF < 5 and Tolerance < 0.2) [41]. Given the large sample size, a more conservative significance criteria of .01 was pre-defined. Furthermore, part correlation which is the correlation between the outcome and the aspect of the predictor unique from all the other predictors, was reported. Thus, the part correlation makes it possible to investigate the relative strengths of the predictors. The strength of the correlation was evaluated according to the following criteria: small = >0.10, medium = >0.30, large = >0.50 [42].

The current sample was matched with the general population of health personnel to ensure that the sample accurately reflected the characteristics of this group. In this population, 15.5% are men and 84% women, and 41.5% are below 39 years of age [43]. In the current sample 84.7% were women and 15.2% were men. However, the sample was somewhat younger than the population, and consequently a sensitivity analysis was performed where the data were stratified for the right percentages of age. The sensitivity analysis yielded identical results to the main analyses, indicating the robustness of the sampling strategy and the presented findings. All analyses were performed using SPSS statistical software version 26.0 [44].

## Results and discussion

### Demographic characteristics

In this epidemiological investigation, 1773 participants were included. Of the participants 178 [10.0%] were medical doctors, 770 [43.4%] were nurses, 244 clinical psychologists [13.7%], and 78 other health workers [4.4%]. Public service providers included social workers (158 [8.9%]), politicians (37 [2.1%]) and other professions (308 [17.4%]). Most of the participants were women (1507 [84.7%]), and had as expected higher education from university (1593 [89.8%], were in married or in a civil union (1193 [67.3%]) and had children (908 [51.2%]), as shown in Table 1.

### Level of PTSD symptoms, anxiety and depression

The levels of PTSD symptoms, anxiety, depression and health anxiety among health personnel and public service providers were high. A total of 28.9% of the sample had clinical or

**Table 1. Demographics and characteristics of the sample.**

| Variable | Frequency, n (%) |
|---|---|
| **Gender** | |
| Male | 269 (15.2) |
| Female | 1502 (84.7) |
| Intersex/transgender | 2 (0.01) |
| **Age** | |
| 18–24 | 242 (13.6) |
| 25–44 | 1054 (59.4) |
| 45–59 | 377 (21.2) |
| >60 | 100 (5.6) |
| **Presence of psychological disorder** | |
| No | 1547 (87.3) |
| Yes | 226 (12.7) |
| **Higher education** | |
| No | 180 (10.1) |
| University | 1593 (89.8) |
| **Married /Civil union** | |
| Yes | 1193 (67.3) |
| No | 580 (32.7) |
| **Children** | |
| Yes | 908 (51.2) |
| No | 865 (48.7) |

subclinical symptoms of PTSD. Furthermore, 21.2% had moderate to severe symptoms of depression and 20.5% had moderate to severe symptoms of anxiety using established cut-offs, as presented in Table 2.

Of those working directly with Covid-19 patients 36.5% had clinical or subclinical symptoms of PTSD in contrast to 27.3% for those working indirectly (see Table 2). A Mann-Whitney U test showed that those working directly in contrast to indirectly with COVID-19 patients had significantly higher PTSD symptoms, $U$ ($N_{direct} = 298$, $N_{indirect} = 1475$) = 183267, $z = -4.62$, $p = <0.001$, and significantly higher depression scores $U$ ($N_{direct} = 298$, $N_{indirect} = 1475$) = 194446, $z = -3.15$, $p = 0.002$. However, there were no significant differences between direct vs. indirect on anxiety $U$ ($N_{direct} = 298$, $N_{indirect} = 1475$) = 207628, $z = -1.51$, $p = 0.130$ and health anxiety, $U$ ($N_{direct} = 298$, $N_{indirect} = 1475$) = 212156, $z = -0.97$, $p = 0.345$.

## Levels of PTSD symptoms among subgroups

Politicians, social workers, and nurses had the highest levels of PTSD symptoms (Table 3). A Kruskal-Wallis H test showed that there was a statistically significant difference in PTSD scores between the different worker groups, $\chi^2$ (6) = 130.3, $p = <0.001$. There were also significant differences between the groups on anxiety $\chi^2$ (6) = 54.6, $p = <0.001$ and depression scores, $\chi^2$ (6) = 103.0, $p = <0.001$, with nurses, social workers and other health workers having the highest levels of symptoms.

## Anxiety and depression as a predictor of PTSD symptoms

A pre-existing psychiatric diagnosis ($p = 0.002$, part correlation = 0.05), higher anxiety ($p = <0.001$, part correlation = 0.25) and higher depression ($p = <0.001$, part correlation = 0.23) was, as hypothesized, associated with higher PTSD symptoms (Table 4).

**Table 2. Cut off scores on PTSD symptoms, anxiety and depression.**

| Scale | Total sample | Working position | | Sex | |
|---|---|---|---|---|---|
| Severity category | | Direct | Indirect | Men | Women |
| **PCL-5, Diagnostic criteria PTSD** | | | | | |
| Non-clinical | 1261 (71.1) | 189 (63.4) | 1072 (72.7) | 215 (79.9) | 1044 (69.5) |
| Subclinical | 305 (17.2) | 67 (22.5) | 238 (16.1) | 38 (14.1) | 267 (17.8) |
| PTSD | 207 (11.7) | 42 (14.1) | 165 (11.2) | 16 (5.9) | 191 (12.7) |
| **PHQ-9, depression symptoms** | | | | | |
| Normal | 774 (43.7) | 106 (35.6) | 668 (45.3) | 159 (59.1) | 613 (40.8) |
| Mild | 624 (35.2) | 114 (38.3) | 510 (34.6) | 84 (31.2) | 540 (36.0) |
| Moderate | 237 (13.4) | 46 (15.4) | 191 (12.9) | 16 (5.9) | 221 (14.7) |
| Moderate severe | 96 (5.4) | 24 (8.1) | 72 (4.9) | 5 (1.9) | 91 (6.1) |
| Severe | 42 (2.4) | 8 (2.7) | 34 (2.3) | 5 (1.9) | 37 (2.5) |
| **GAD-7, anxiety symptoms** | | | | | |
| Normal | 963 (54.3) | 152 (51.0) | 811 (55.0) | 189 (70.3) | 773 (51.5) |
| Mild | 446 (25.2) | 75 (25.2) | 371 (25.2) | 49 (18.2) | 396 (26.4) |
| Moderate | 286 (16.1) | 53 (17.8) | 233 (15.8) | 22 (8.2) | 264 (17.6) |
| Severe | 78 (4.4) | 18 (6.0) | 60 (4.1) | 9 (3.3) | 69 (4.6) |

*Note*: PCL-5 used 31> as a cut-off for PTSD. Diagnostic criterion was based on DSM-5. Percentages in parenthesis.

## Other predictors of PTSD symptoms

Worries about job and economy ($p = <0.001$, part correlation = 0.07), negative metacognitions ($p = <0.001$, part correlation = 0.09), burnout ($p = 0.001$, part correlation = 0.05) and health anxiety ($p = <0.001$, part correlation = 0.10), and emotional support ($p = 0.007$, part correlation = -0.04 were significantly associated with PTSD symptoms. After controlling for demographic variables, anxiety and depression, and working directly vs. indirect with Covid-19 patients, interpersonal problems ($p = 0.015$, part correlation = 0.04), and positive metacognitions ($p = 0.011$, part correlation = 0.04) were not associated with PTSD symptoms, as reported in Table 5.

## Discussion

The current epidemiological investigation, among 1773 health personnel and public service providers, reveals a high point-prevalence of PTSD symptoms (28.9%), anxiety (20.5%) and

**Table 3. Scores of post-traumatic symptoms, anxiety, depression and health anxiety in total sample and subgroups.**

| Sample | PCL-5 Median (IQR) | GAD-7 Median (IQR) | PHQ-9 Median (IQR) | Health anxiety Median (IQR) |
|---|---|---|---|---|
| Total sample (N = 1773) | 8 (2.0–19.0) | 4 (2.0–7.0) | 5 (3.0–9.0) | 1 (0–3.0) |
| Direct Covid-19 (n = 298) | 11 (4.0–26.3) | 4 (2.0–7.0) | 6 (3.0–10.0) | 1 (0.8–3.0) |
| Indirect Covid-19 (n = 1475) | 8 (2.0–18.0) | 4 (2.0–7.0) | 5 (3.0–9.0) | 1 (0–3.0) |
| Medical doctors (n = 178) | 5 (1.0–14.0) | 3 (1.0–6.0) | 3 (2.0–6.0) | 1 (0–2.0) |
| Nurses (n = 770) | 10 (3.0–24.0) | 5 (2.0–8.0) | 6 (3.0–10.0) | 2 (1.0–3.0) |
| Clinical Psychologists (n = 244) | 3 (0–8.0) | 3 (1.0–6.0) | 4 (2.0–6.0) | 1 (0–2.0) |
| Social workers (n = 158) | 11 (4.0–23.0) | 5 (2.0–7.0) | 6 (4.0–11.0) | 2 (1.0–3.0) |
| Politicians (n = 37) | 14 (3.5–24.5) | 3 (1.0–7.5) | 5 (3.0–8.0) | 2 (0–4.5) |
| Other health workers (n = 78) | 9 (4.8–20.3) | 5 (2.0–7.0) | 6 (3.0–9.0) | 2 (1.0–3.0) |
| Other (n = 308) | 8 (2.3–18.8) | 4 (2.0–7.0) | 5 (3.0–9.0) | 1 (0–3.0) |

*Note*: Other included health personnel working in other sectors or public service providers not working as social workers.

**Table 4. Anxiety, depression, and diagnosis as predictor for PTSD symptoms.**

|  | Unstandardized regression coefficient | Standard Error | T-value | p | Part |
|---|---|---|---|---|---|
| Predictors of PTSD symptoms (PCL-5), $N$ = 1773, Adjusted R$^2$ = 0.47 |  |  |  |  |  |
| Intercept | 0.34 | 0.39 | 0.86 | .392 |  |
| Diagnosis | 2.32 | 0.76 | 3.03 | .002 | 0.05 |
| Anxiety | 1.19 | 0.08 | 14.21 | < .001 | 0.25 |
| Depression | 0.96 | 0.07 | 13.21 | < .001 | 0.23 |

depression (21.2%). The results demonstrate levels of PTSD that have been reported among health personnel in other surveys, but are considerably higher than a recent COVID-19 study from Singapore [11–15]. However, exact comparisons are made difficult by the fact that some of the other studies have used different measures and not consistently reported sub-clinical symptoms. The current study used PCL-5, which fully aligns to DSM-V criteria. However, in a recent study, PTSD symptoms among health-workers in India and Singapore during the COVID-19 outbreak the prevalence-rate of 9.3% experiencing PTSD-symptoms, were lower than in the current study [45]. There could be several reasons for differences in prevalence-rates between countries and firm conclusions are hampered by the use of different measures to assess PTSD symptoms. It is suggested, however, that that some western countries have higher risk of PTSD because there are high expectations for risk-fee life and high attention to potential harmful mental health effects of serious life events [46].

Health personnel and public service providers working directly with COVID-19 patients reported more severe symptoms of PTSD and depression. The results mirror the findings in similar COVID-19 studies from China [6] and other pandemics [4, 5], where health workers working in the frontline of pandemics reported higher levels of distress. A large proportion of the respondents had subclinical PTSD symptoms, 22.5% of those working directly and 16.1% of those working indirectly. Those having subclinical symptoms are vulnerable to developing clinical PTSD, especially because the work situation will constantly pose challenges and

**Table 5. Risk factors for PTSD symptoms identified by multiple regression analysis.**

|  | Unstandardized regression coefficient | Standard Error | T-value | p | Part |
|---|---|---|---|---|---|
| Predictors of PTSD symptoms (PCL-5), $N$ = 1771, Adjusted R$^2$ = 0.53 |  |  |  |  |  |
| Intercept | -4.66 | 1.55 | -3.01 | < .001 |  |
| Indirect vs. direct | 2.28 | 0.60 | 3.79 | < .001 | 0.06 |
| Diagnosis | 1.22 | 0.73 | 1.67 | .095 | 0.03 |
| Age category | 1.73 | 0.32 | 5.44 | < .001 | 0.09 |
| Gender | -1.13 | 0.64 | -1.76 | .079 | -0.03 |
| Relationship | 0.30 | 0.52 | 0.58 | .564 | <0.01 |
| Children | -1.18 | 0.49 | -2.37 | .018 | -0.04 |
| Depression | 0.67 | 0.08 | 8.23 | < .001 | 0.14 |
| Anxiety | 0.76 | 0.09 | 8.48 | < .001 | 0.14 |
| Emotional support | -0.33 | 0.12 | -2.71 | .007 | -0.04 |
| Worry job/eco. | 0.76 | 0.18 | 4.32 | < .001 | 0.07 |
| Health anxiety | 0.69 | 0.12 | 5.80 | < .001 | 0.10 |
| Burnout | 1.02 | 0.31 | 3.27 | .002 | 0.05 |
| Interpersonal prob. | 0.08 | 0.03 | 2.43 | .015 | 0.04 |
| Metacog. pos | 0.008 | 0.003 | 2.55 | .011 | 0.04 |
| Metacog. neg | 0.020 | 0.004 | 5.41 | < .001 | 0.09 |

stressful incidents. Hence, working directly or indirectly with COVID-19 patients should be regarded as a risk factor for developing PTSD symptoms, thus underlining the importance of monitoring the subclinical symptoms among individuals working with COVID-19 patients. Moreover, the findings revealed that occupations other than nurses and medical doctors are also highly affected by the pandemic, especially social workers, other health workers and politicians.

Predictors of PTSD are of importance to identify those who may be at risk of developing PTSD. Having a pre-existing psychiatric diagnosis, higher levels of anxiety, and depression symptoms were associated with more PTSD symptoms, which is in accordance with previous findings in the literature [21]. Thus, anxiety and depression symptoms may be a source of vulnerability to developing PTSD symptoms during pandemics. Among state predictors relevant as possible targets for intervention, worries about job and economy were significant, highlighting the importance of these worry themes in association with PTSD symptoms. Worry is a central maintaining factor in psychopathology [28]. Thus, governments may try to take specific actions to reduce worries during pandemics, which may be achieved by providing accurate information about viral transmission chains, and reducing uncertainty about jobs and the economy if possible. A recent study on the COVID-19 pandemic revealed the beneficial impact of sufficient information on depression and anxiety [30]. Health anxiety was also significantly associated with PTSD symptoms, which may indicate the importance of also addressing worry about health.

Burnout was positively associated with PTSD symptoms. Previous findings have revealed that health workers have high levels of stress [47] which may lead to burnout, highlighting the importance of investigating the associations between stress, burnout and PTSD symptoms.

Negative metacognitions, but not positive, predicted PTSD symptoms, which is in accordance with the previous literature [27, 28]. The patients' negative thoughts about their own thinking, such as "I cannot control my thinking" are an important variable related to PTSD symptoms. Reducing dysfunctional metacognitions and increasing participants' ability to reduce worry and rumination may therefore be an important asset during pandemics, and negative metacognitions should be further investigated as a possible treatment-target [28].

Interpersonal problems was not significantly associated with PTSD symptoms, however emotional support was, but the part correlations were low, which is not in accordance with the hypothesis as outlined [18, 25]. The numerous significant predictors show that there may be many routes to reducing PTSD symptoms among health workers and public service providers. However, health anxiety, anxiety and depression had clinically relevant effect sizes = >0.1 as measured by part correlation. The other predictors had smaller effect sizes, which may be less relevant in terms of clinical significance.

Protecting government officials and politicians against symptoms of anxiety and depression are imperative, as these disorders are associated with reduced cognitive capacities [48], which may lead to unfavorable consequences in pandemics where the pressure to make decision is already high given the intensity of pandemic incidents. Similarly, doctors and nurses often have to make decisions about life and death, decisions requiring cognitive capacities which may be burdened by symptoms of depression and anxiety. Similarly public service providers are of critical importance during a pandemic, and care must be to ensure that they receive the best treatment possible.

As requested in several papers, some form of psychological first aid could be needed to help healthcare workers [2, 29]. Specific action on negative metacognition, worry, anxiety, and depression may be a pathway forward to reducing PTSD symptoms. However, studies with repeated measurements are needed to assess temporal precedence, which is important for identifying causal relations and inform treatments.

## Strengths and limitations

Strengths of this study include that it captured the effects of NPIs momentarily as they happened and were held constant during the measurement period. Given that the NIPs were globally implemented, the results in this study is probably generalizable to other countries. Limitations include that burnout was measured by only one item, making it prone to measurement error. The sample was cross-sectional which precludes conclusions about causality. The group public service providers had lack of diversity consisting of only social workers and politicians reducing the possibility to generalize across this group. The sample was somewhat biased on age. However, the main findings were replicated with a sensitivity analysis of a randomly selected subset of individuals with demographic characteristics accurately matching population parameters, further attesting to the robustness of the presented results.

## Conclusions

Health workers and public service providers have markedly high levels of PTSD symptoms, anxiety and depression during the COVID-19 pandemic. Those working directly with COVID-19 patients have significantly higher levels of PTSD symptoms and depression compared to those working indirectly. These increases in symptoms are markedly higher than estimates from pre-pandemic populations, suggesting that this issue may be a major cause for concern. Special care should be taken to assess the level of PTSD symptoms among both health personnel and public service providers in the forthcoming period. Appropriate action to monitor and reduce PTSD, anxiety, and depression among these groups of individuals working in the frontline of pandemic with crucial societal roles should be taken immediately.

## Supporting information

**S1 File. Overview of questionnaires and items.**
(DOCX)

## Acknowledgments

We thank all the respondents that participated in the study.

## Author Contributions

**Conceptualization:** Sverre Urnes Johnson, Omid V. Ebrahimi, Asle Hoffart.

**Data curation:** Sverre Urnes Johnson, Omid V. Ebrahimi, Asle Hoffart.

**Formal analysis:** Sverre Urnes Johnson, Asle Hoffart.

**Investigation:** Sverre Urnes Johnson, Omid V. Ebrahimi, Asle Hoffart.

**Methodology:** Sverre Urnes Johnson, Omid V. Ebrahimi, Asle Hoffart.

**Project administration:** Sverre Urnes Johnson, Omid V. Ebrahimi.

**Software:** Sverre Urnes Johnson, Omid V. Ebrahimi.

**Validation:** Sverre Urnes Johnson.

**Writing – original draft:** Sverre Urnes Johnson.

**Writing – review & editing:** Sverre Urnes Johnson, Omid V. Ebrahimi, Asle Hoffart.

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
