## [Decision Letter · Decision Letter 0]

24 Jul 2020

PONE-D-20-19419

PTSD-symptoms Among Health Workers and Public Service Providers During the COVID-19 Outbreak

PLOS ONE

Dear Dr. Johnson,

Thank you for submitting your manuscript to PLOS ONE. After careful consideration, we feel that it has merit but does not fully meet PLOS ONE’s publication criteria as it currently stands. Therefore, we invite you to submit a revised version of the manuscript that addresses the points raised during the review process.

The submission received three thoughtful reviews yielding three different decisions (Accept, Minor Revision and Major Revision). After carefully considering the reviews, and my own thoughts, I am making a decision of Minor Revision. This decision to me reflects the essence of the reviewers' comments.  Specifically, all reviewers noted strengths of this work, but varied in the extent to which they had suggestions for revision. The Major Revision decision (Reviewer 3) includes important suggestions, but addressing the feedback of Reviewer 3 will not require extensive re-writing or numerous data analyses. Even the reviewer who offered an Accept decision did pose a suggestion (to make the data more available if possible). 

I will not repeat all of the reviewers' comments here, but ask you to attend carefully to each suggestion. In your revision, please explain how the comments were addressed. If you decide not to action a particular comment, please explain why. Two of the three reviewers (Reviewers 2 and 3) both asked about the availability of the data. I understand the grave importance of following the ethics board's rules, but data availability is extremely important. If you would please consider trying to find a way to enable access to the de-identified data without restriction (after consultation with the ethics board), that would be appreciated.  I look forward to receiving your revision. Thank you again for submitting this research to PLOS ONE.

We look forward to receiving your revised manuscript.

Kind regards,

Kristin Vickers, Ph.D.

Academic Editor

PLOS ONE

Additional Editor Comments:

Dear Dr. Johnson,

Thank you for submitting your timely research (“PTSD-symptoms Among Health Workers and Public Service Providers During the COVID-19 Outbreak”) to PLoS ONE. The submission received three thoughtful reviews yielding three different decisions (Accept, Minor Revision and Major Revision). After carefully considering the reviews, and my own thoughts, I am making a decision of Minor Revision. This decision to me reflects the essence of the reviewers' comments. Specifically, all reviewers noted strengths of this work, but varied in the extent to which they had suggestions for revision. The Major Revision decision (Reviewer 3) includes important suggestions, but addressing the feedback of Reviewer 3 will not require extensive re-writing or numerous data analyses. Even the reviewer who offered an Accept decision did pose a suggestion (to make the data more available if possible).

I will not repeat all of the reviewers' comments here, but ask you to attend carefully to each suggestion. In your revision, please explain how the comments were addressed. If you decide not to action a particular comment, please explain why. Two of the three reviewers (Reviewers 2 and 3) both asked about the availability of the data. I understand the grave importance of following the ethics board's rules, but data availability is extremely important. If you would please consider finding a way to enable access to the de-identified data without restriction (after consultation with the ethics board), that would be appreciated. Please note the limitations that Reviewers 1 and 3 each suggested, as well as the additional information that each felt would be important to include. I look forward to receiving your revision. Thank you again for submitting this research to PLos ONE.

Journal Requirements:

3. Please upload a copy of the Supporting Information which you refer to in your text on page 8.

Reviewers' comments:

Reviewer's Responses to Questions

**Comments to the Author**

1. Is the manuscript technically sound, and do the data support the conclusions?

Reviewer #1: Yes

Reviewer #2: Yes

Reviewer #3: Partly

2. Has the statistical analysis been performed appropriately and rigorously? 

Reviewer #1: Yes

Reviewer #2: Yes

Reviewer #3: Yes

3. Have the authors made all data underlying the findings in their manuscript fully available?

Reviewer #1: Yes

Reviewer #2: No

Reviewer #3: No

4. Is the manuscript presented in an intelligible fashion and written in standard English?

Reviewer #1: Yes

Reviewer #2: Yes

Reviewer #3: Yes

5. Review Comments to the Author

Reviewer #1: The authors present findings from a large cohort size, reporting the prevalence of PTSD symptoms amongst their cohort. This manuscript may not be novel, but it does add to the body of literature to help understand the psychological impact of covid both in Norway and Globally. The overall methodology and analyses is robust. A few minor amendments is suggested.

1. In discussion, please attempt to compare PTSD symptom prevalence to other countries and discuss accordingly: e.g. Chew NW, Lee GK, et al. A multinational, multicentre study on the psychological outcomes and associated physical symptoms amongst healthcare workers during COVID-19 outbreak. Brain, behavior, and immunity. 2020 Apr 21.

---Although other papers may use different instruments to measure PTSD and this is an inherent limitation.

--did factors like case volume in the country, healthcare resources strain, mortality rate play a role?

2. one major limitation is that it is a cross-sectional study. we are unsure how much of this is related to covid directly, or may have been pre-existing prior to covid. the authors should include this as a limitation. further longitudinal study in a similar cohort can also be used to assess the long-term PTSD impact once the pandemic blows over.

Reviewer #2: The authors present a timely examination of PTSD, depression, and anxiety in relation to COVID-19 pandemic in a large sample of healthcare providers out of Norway. While this is a large study with self-report data, the timeliness of the topic combined with efficient and well-done study design (in choice of measures, etc.) make this manuscript a useful contribution to the literature as we try to figure out what needs to be done to reduce the mental health impacts of the current pandemic.

Introduction is concise and well written to support the design.

Methods are good and results clearly presented.

Discussion is well reasoned, sticks to the results presented and provides a key contribution to the literature.

Given the journal requirements to provide data or a rationale why data cannot be provided, I believe additional rationale beyond ethics needs to be provided, but that is my only hesitation.

Reviewer #3: Thank you for the opportunity to review this manuscript. The authors conducted a timely examination of self-reported symptoms for front-line workers during the current global pandemic. The novelty of examining public service workers is unique to the study, but not as distinguished in both the methodology nor discussion. Rather, the study methodologies, in particular, the recruitment of target populations was not supported by, or in line with the strong theoretical rationale. Below, I outline some of my thoughts.

Background and introduction:

1. Background is well-written, with strong rationale in conducting this research.

Methods:

1. Some clarifications are needed as there seems to be a disconnect between the background rationale and study methodology. While the examination of healthcare and public service providers are clear from the background and rationale, the exact recruitment and populations reached are not. For example, which groups were considered healthcare professional – those working in ER/COVID units? Nurses in long-term homes? What about other healthcare professionals, and particularly for elective procedures, like mental health professionals, family physician, physiotherapists, pharmacists, etc.

2. Similarly, for public service professionals, what is considered "public service"? What about essential workers such as bus drivers, grocery store clerks, who likely have less protection compared to healthcare professionals, earn a lower wage, and are still required to work during the pandemic? Are they factored into your recruitment strategy?

3. The recruitment strategy is missing some details. For example, what did the Facebook algorithm determine for health personnel and public service providers, and who were the politicians reached?

4. For public service providers, there was only options for social workers and politicians. This is not reflective of the diversity of public service providers nor exhaustive enough to draw conclusions from.

5. Demographic variables did not consider contact and exposure to covid-19, which may be central to this line of research.

6. Please clarify experimental groupings and rationale. Which groups/professions were considered healthcare workers vs. public service workers. These were not mentioned until demographic characteristics in the results section.

7. I was not able to locate the supplementary file for the single item measure of burnout. How was this construct quantified/operationalized? If it’s a single item, please insert item right into text.

Results:

1. Unclear about direct vs indirect work relationships with COVID-19 from both methodology and results. How was this operationalized, was frequency of contact, size of hospital/healthcare institution, presence of outbreaks, and other variables considered?

2. Please fix "other predictors of treatment outcome". This is a cross-sectional inferential analysis, no treatment was offered.

Discussion:

1. Given the focus of the discussion on distinctions between public service vs. healthcare professionals, it is important to clearly operationalize and provide clear details about recruitment strategies and conditions for inclusion/exclusion.

2. With small proportion of public service professionals sampled, and lack of clarify regarding the types and diversity of profession, I caution the authors in drawing such strong conclusions from data collected. Would recommend to highlight as a limitation in data interpretations.

6. PLOS authors have the option to publish the peer review history of their article (what does this mean?). If published, this will include your full peer review and any attached files.

Reviewer #1: No

Reviewer #2: No

Reviewer #3: No

---

## [Author Response · Author response to Decision Letter 0]

27 Aug 2020

Response to Reviewers

PONE-D-20-19419

PTSD-symptoms Among Health Workers and Public Service Providers During the COVID-19 Outbreak

We would like to thank the reviewers and editorial team for their comments on the paper, which gave us the opportunity to make improvements in the manuscript. Detailed responses to reviewers’ comments are given for each point separately. Each response is identified by the number of the reviewers and the number of the comment. Our responses are given below the comment.

The PLOS ONE style requirements have been revisited, and the paper should currently meet these requirements. 

As requested by the editor I have been in contact with NSD and REK.

The present dataset includes information about specific behaviors, trait variables, and health outcomes of a large and representative sample of the adults. In accordance with the guidelines of The Centre for Research Data (NSD) and The Regional Committee for Medical and Health Research Ethics (REK), access to the data can only be provided to qualified investigators whose proposed use of the data have been approved by all relevant independent review committees in the conducted countries of the study, including REK and NSD, and whose research plans provide a defensible proposal and methodologically sound design with the aims approved by REK, NSD, the university ethics board, and other necessary organizations that require an application process for providing access to sensitive data containing predictive information on a large and representative sample of the adult population in the sample country.

The main reasons for the constrain on the use of the data is the following. First, free use of the data, even in anonymized form, is not mentioned in the letter of informed consent, which is a prerequisite for the study. Second, the deposit of anonymized data, on an international server is not allowed according the ethical board based on our informed consent. However as specified, the data can be requested from the first author following ethical approval of access to the data from NSD and REK. REK and NSD does not handle request for the use of data on behalf of the author. 

We suggest the following data availability statement which is in accordance with other papers from Norway published in PLOS ONE: 

Suggestion for Data Availability Statement:

 Our ethical approval granted by the Regional Committees for Medical and Health Research Ethics in Norway does not allow us to submit the data to a Public repository. In line with the ethics approval, the data are to be kept at a secure server only accessible by the authors at the University of Oslo. Access to the data can be granted from the first author following ethical approval of suggested project plan for the use of data from NSD and REK. Such requests are to be sent to Associate Professor, Sverre Urnes Johnson, Department of Psychology, University of Oslo, Forskningsveien 3 A, 0373 Oslo, Norway, Email: s.u.johnson@psykologi.uio.no, phone: +47-22845295

3. Please upload a copy of the Supporting Information which you refer to in your text on page 

A copy of the supplementary material encompassing all the different items used in the study is now uploaded. 

Reviewer #1:

The authors present findings from a large cohort size, reporting the prevalence of PTSD symptoms amongst their cohort. This manuscript may not be novel, but it does add to the body of literature to help understand the psychological impact of covid both in Norway and Globally. The overall methodology and analyses is robust. A few minor amendments is suggested.

1. In discussion, please attempt to compare PTSD symptom prevalence to other countries and discuss accordingly: e.g. Chew NW, Lee GK, et al. A multinational, multicentre study on the psychological outcomes and associated physical symptoms amongst healthcare workers during COVID-19 outbreak. Brain, behavior, and immunity. 2020 Apr 21.

The paper by Chew et al is included in the revised manuscript. 

The results from the current paper is discussed in comparison with other studies, including the study from Chew et al: See page 12: “However in a recent study, PTSD symptoms among health-workers in India and Singapore during the COVID-19 outbreak the prevalence-rate of 9.3% experiencing PTSD-symptoms, were lower than in the current study [43]. There could be several reasons for differences in prevalence-rates between countries and firm conclusions are hampered by the use of different measures to assess PTSD symptoms. It is suggested, however, that that some western countries have higher risk of PTSD because there are high expectations for risk-fee life and high attention to potential harmful mental health effects of serious life events [44]”. 

-Although other papers may use different instruments to measure PTSD and this is an inherent limitation.

This is a clear limitation, and is mentioned in the discussion, see page 12.

--did factors like case volume in the country, healthcare resources strain, mortality rate play a role?

Thank you for a good suggestion. Norway did not have extraordinary many deaths related to Covid-19 and normally you would expect that more strain on healthcare resources and higher mortality rate would be associated with higher PTSD-symptoms, but this does not seem to be the case. India have for example higher mortality rate compared to Norway (https://coronavirus.jhu.edu/data/mortality). Thus, the paper by Heir et al about possible higher prevalence of PTSD in the certain western populations is relevant and mentioned in the discussion (page 12).

2. one major limitation is that it is a cross-sectional study. we are unsure how much of this is related to covid directly, or may have been pre-existing prior to covid. the authors should include this as a limitation. further longitudinal study in a similar cohort can also be used to assess the long-term PTSD impact once the pandemic blows over.

We have highlighted that the cross-sectional methodology hinders causal conclusion, see page 15. Further longitudinal studies of the same cohort are planned.

Reviewer #2: The authors present a timely examination of PTSD, depression, and anxiety in relation to COVID-19 pandemic in a large sample of healthcare providers out of Norway. While this is a large study with self-report data, the timeliness of the topic combined with efficient and well-done study design (in choice of measures, etc.) make this manuscript a useful contribution to the literature as we try to figure out what needs to be done to reduce the mental health impacts of the current pandemic.

Introduction is concise and well written to support the design.

Methods are good and results clearly presented.

Discussion is well reasoned, sticks to the results presented and provides a key contribution to the literature.

Thank you for the positive feedback.

Given the journal requirements to provide data or a rationale why data cannot be provided, I believe additional rationale beyond ethics needs to be provided, but that is my only hesitation.

The reason for unavailability of sharing through a deposit of anonymized data is given in the beginning of this document.

Reviewer #3: Thank you for the opportunity to review this manuscript. The authors conducted a timely examination of self-reported symptoms for front-line workers during the current global pandemic. The novelty of examining public service workers is unique to the study, but not as distinguished in both the methodology nor discussion. Rather, the study methodologies, in particular, the recruitment of target populations was not supported by, or in line with the strong theoretical rationale. Below, I outline some of my thoughts.

Background and introduction:

1. Background is well-written, with strong rationale in conducting this research.

Thank you

Methods:

1. Some clarifications are needed as there seems to be a disconnect between the background rationale and study methodology. While the examination of healthcare and public service providers are clear from the background and rationale, the exact recruitment and populations reached are not. For example, which groups were considered healthcare professional – those working in ER/COVID units? Nurses in long-term homes? What about other healthcare professionals, and particularly for elective procedures, like mental health professionals, family physician, physiotherapists, pharmacists, etc.

We have created at new paragraph, where we explain in further details the recruitment procedure. “The participants were either health personnel or public service providers working directly or indirectly with COVID-19 patients. The following groups of health personnel were assessed: Medical doctors, nurses, clinical psychologist and other health workers (not specified). The following group of public service providers working with consequences of the COVID-19 pandemic were assessed and grouped: social workers, politicians and other professions (not specified).”

 As specified we did not assess the level of detail as the reviewer suggest, thus a category of “other” exist both for health personnel and public service providers. 

2. Similarly, for public service professionals, what is considered "public service"? What about essential workers such as bus drivers, grocery store clerks, who likely have less protection compared to healthcare professionals, earn a lower wage, and are still required to work during the pandemic? Are they factored into your recruitment strategy?

Public service is a broad term, but in the questionnaire the work had to be related to consequences of the pandemic and those answering as public service providers worked indirectly with the consequences. However, the questionnaire was not detailed enough to separate between grocery store clerks, bus drivers etc. We have specified the term public service provider, see page 7: “The following group of public service providers working with consequences of the COVID-19 pandemic were assessed and grouped: social workers, politicians and other professions (not specified)”.

3. The recruitment strategy is missing some details. For example, what did the Facebook algorithm determine for health personnel and public service providers, and who were the politicians reached?

 The Facebook algorithm reaches a random sample of individuals including and above 18 years of age who have reported their labor to fit our target categories, such as nurses, doctors, psychologists, and other individuals working in the health-care system, in addition to those reported to be politicians and social workers. The algorithm is inherently designed to optimize for genuine human activity and uses a variety of methods to remove false and duplicate accounts from the selection algorithm. Upon taking the survey, these participants identify and register themselves through a platform referred to as Services for Sensitive Data (TSD), where their information is safely stored. With the imputed parameters, the algorithm reached a total of 12 113 individuals meeting the aforementioned criteria.

4. For public service providers, there was only options for social workers and politicians. This is not reflective of the diversity of public service providers nor exhaustive enough to draw conclusions from.

This is a relevant point, and the lack of diversity of the construct “public service providers” is now mentioned as a limitation at page 15: “The group public service providers had lack of diversity consisting of only social workers and politicians reducing the possibility to generalize across this group.”

5. Demographic variables did not consider contact and exposure to covid-19, which may be central to this line of research.

We did have questions about contact and exposure to Covid-19 and whether the person was in quarantine, but we did not use the variables in this paper, because we think the variables are less central for the questions asked in this study and the variables were used in another publication:

 Ebrahimi OV, Hoffart A, Johnson SU. The mental health impact of non-pharmacological interventions aimed at impeding viral transmission during the COVID-19 pandemic in a general adult population and the factors associated with adherence to these mitigation strategies. 2020, May 9. doi: https://doi.org/10.31234/osf.io/kjzsp.

6. Please clarify experimental groupings and rationale. Which groups/professions were considered healthcare workers vs. public service workers. These were not mentioned until demographic characteristics in the results section.

The groups are clarified and mentioned below the heading “participants”, see page 7: 

“The participants were either health personnel or public service providers working directly or indirectly with COVID-19 patients. The following groups of health personnel were assessed: Medical doctors, nurses, clinical psychologist and other health workers (not specified). The following group of public service providers working with consequences of the COVID-19 pandemic were assessed and grouped: social workers, politicians and other professions (not specified).”

7. I was not able to locate the supplementary file for the single item measure of burnout. How was this construct quantified/operationalized? If it’s a single item, please insert item right into text.

A supplementary file consisting of all the questionnaires used in the study is uploaded. Furthermore, the single items “I feel burned out” is inserted in the text.

Results:

1. Unclear about direct vs indirect work relationships with COVID-19 from both methodology and results. How was this operationalized, was frequency of contact, size of hospital/healthcare institution, presence of outbreaks, and other variables considered?

As specified on page 5 at the end of the introduction: 

Directly is defined as face-to-face contact with patients that have tested positive for COVID-19. Indirectly is defined as working with other consequences of the COVID-19 pandemic, but not face-to-face with patients that is infected with COVID-19.

2. Please fix "other predictors of treatment outcome". This is a cross-sectional inferential analysis, no treatment was offered.

The sentence is changed to “Other predictors of PTSD symptoms”.

Discussion:

1. Given the focus of the discussion on distinctions between public service vs. healthcare professionals, it is important to clearly operationalize and provide clear details about recruitment strategies and conditions for inclusion/exclusion.

We have tried to make the information about the two groups clearer throughout the manuscript. 

2. With small proportion of public service professionals sampled, and lack of clarify regarding the types and diversity of profession, I caution the authors in drawing such strong conclusions from data collected. Would recommend to highlight as a limitation in data interpretations.

We have highlighted the heterogeneity of the group “public service providers” as a limitation, see page 15: “The group public service providers had lack of diversity consisting of only social workers and politicians reducing the possibility to generalize across this group.”

---

## [Decision Letter · Decision Letter 1]

30 Sep 2020

PONE-D-20-19419R1

PTSD-symptoms Among Health Workers and Public Service Providers During the COVID-19 Outbreak

PLOS ONE

Dear Dr. Johnson,

Thank you for submitting your manuscript to PLOS ONE. After careful consideration, we feel that it has merit but does not fully meet PLOS ONE’s publication criteria as it currently stands. Therefore, we invite you to submit a revised version of the manuscript that addresses the points raised during the review process.

We look forward to receiving your revised manuscript.

Kind regards,

Kristin Vickers, Ph.D.

Academic Editor

PLOS ONE

Additional Editor Comments:

Dear Dr. Johnson,

Reviewers were very satisfied with your revisions; thank you for your efforts. The reason for the Minor Revision decision is that I need to ask you to clarify the extent to which there is overlap (if any) between this manuscript and the work by Ebrahimi et al. Specifically, in the Response to Reviewers document, the following is noted:

"We did have questions about contact and exposure to Covid-19 and whether the person was in quarantine, but we did not use the variables in this paper, because we think the variables are less central for the questions asked in this study and the variables were used in another publication:

Ebrahimi OV, Hoffart A, Johnson SU. The mental health impact of nonpharmacological interventions aimed at impeding viral transmission during the COVID-9 pandemic in a general adult population and the factors associated with adherence to these mitigation strategies. 2020, May 9. doi: " ext-link-type="uri" xlink:type="simple">https://doi.org/10.31234/osf.io/kjzsp."

PLOS ONE has guidelines when different publications are related. If there is no overlap between this manuscript and the other publication (Ebrahimi et al.), please let me know that. If there is any overlap, please follow the PLOS ONE guidelines at

https://journals.plos.org/plosone/s/submission-guidelines#loc-related-manuscripts

Specifically, upon your submission of your revised manuscript, please indicate whether there are any related manuscripts under consideration or published elsewhere. If related work has been submitted or published elsewhere, please include a copy of it with your revised manuscript and describe its relation to the submitted work. I will also ask that that the authors make it clear in the revised manuscript that results from a related dataset have been published previously or are under consideration for publication, if applicable.

Thank you for submitting your research to PLOS ONE and I look forward to receiving your revision. Please let me know if you have any questions.

Reviewers' comments:

Reviewer's Responses to Questions

**Comments to the Author**

1. If the authors have adequately addressed your comments raised in a previous round of review and you feel that this manuscript is now acceptable for publication, you may indicate that here to bypass the “Comments to the Author” section, enter your conflict of interest statement in the “Confidential to Editor” section, and submit your "Accept" recommendation.

Reviewer #1: All comments have been addressed

Reviewer #2: All comments have been addressed

Reviewer #3: All comments have been addressed

2. Is the manuscript technically sound, and do the data support the conclusions?

Reviewer #1: Yes

Reviewer #2: Yes

Reviewer #3: Yes

3. Has the statistical analysis been performed appropriately and rigorously? 

Reviewer #1: Yes

Reviewer #2: Yes

Reviewer #3: Yes

4. Have the authors made all data underlying the findings in their manuscript fully available?

Reviewer #1: Yes

Reviewer #2: Yes

Reviewer #3: No

5. Is the manuscript presented in an intelligible fashion and written in standard English?

Reviewer #1: Yes

Reviewer #2: Yes

Reviewer #3: Yes

6. Review Comments to the Author

Reviewer #1: The authors have responded appropriately to the comments. I have no further comments, and I feel the manuscript is now suitable for publication.

Reviewer #2: All reviewer comments have been adequately addressed. The bypass was not working for me to bypass entering all fields.

Reviewer #3: The authors have addressed my feedback. This will make a timely contribution to our understanding of the literature surrounding global COVID19 responses.

7. PLOS authors have the option to publish the peer review history of their article (what does this mean?). If published, this will include your full peer review and any attached files.

Reviewer #1: No

Reviewer #2: No

Reviewer #3: No

---

## [Author Response · Author response to Decision Letter 1]

5 Oct 2020

Revision letter 5.10.2020

PONE-D-20-19419

PTSD-symptoms Among Health Workers and Public Service Providers During the COVID-19 Outbreak

We would like to thank the reviewers and editorial team for their comments on related manuscripts. Detailed responses to editor comments (E) are given for each point separately, behind the letter A (authors). 

E1: PLOS ONE has guidelines when different publications are related. If there is no overlap between this manuscript and the other publication (Ebrahimi et al.), please let me know that. If there is any overlap, please follow the PLOS ONE guidelines at

https://journals.plos.org/plosone/s/submission-guidelines#loc-related-manuscripts

A: It is unclear for the authors how PLOS ONE defines “overlap”, but for the sake of transparency we have given a statement about any potential overlap. 

E2: Specifically, upon your submission of your revised manuscript, please indicate whether there are any related manuscripts under consideration or published elsewhere. If related work has been submitted or published elsewhere, please include a copy of it with your revised manuscript and describe its relation to the submitted work. 

A2: We have included a copy of three separate papers from the same dataset which have been submitted for publication. We don`t think that there are any substantial overlap since the first, Ebrahimi et al, targets the general population for anxiety and depression, Hoffart et al., targets loneliness in the general population and Johnson et al., targets parental stress in the general population. However, the data from the same data-collection.

E3: I will also ask that that the authors make it clear in the revised manuscript that results from a related dataset have been published previously or are under consideration for publication, if applicable.

A3: In the revised manuscript the following information is added: “Articles from the same project, but with different topics, concerning the prevalence of anxiety and depression [30], loneliness [31]and parental stress [32] are under consideration for publication.”

---

## [Editor Report · Decision Letter 2]

8 Oct 2020

PTSD-symptoms Among Health Workers and Public Service Providers During the COVID-19 Outbreak

PONE-D-20-19419R2

Dear Dr. Johnson,

We’re pleased to inform you that your manuscript has been judged scientifically suitable for publication and will be formally accepted for publication once it meets all outstanding technical requirements.

Kind regards,

Kristin Vickers, Ph.D.

Academic Editor

PLOS ONE

Additional Editor Comments (optional):

Thank you to the authors for fully explaining the nature of the different papers coming from this project. I also appreciate that the authors re-did analyses. This is timely and important research.

---

## [Editor Report · Acceptance letter]

14 Oct 2020

PONE-D-20-19419R2 

PTSD symptoms among health workers and publicservice providers during the COVID-19 outbreak 

Dear Dr. Johnson:

I'm pleased to inform you that your manuscript has been deemed suitable for publication in PLOS ONE. Congratulations! Your manuscript is now with our production department. 

Kind regards, 

on behalf of

Dr. Kristin Vickers 

Academic Editor

PLOS ONE